# EVALUATION METHODOLOGY FOR ATTACKS AGAINST CONFIDENCE THRESHOLDING MODELS

## ABSTRACT

Current machine learning algorithms can be easily fooled by **adversarial examples**. One possible solution path is to make models that use **confidence thresholding** to avoid making mistakes. Such models refuse to make a prediction when they are not confident of their answer. We propose to evaluate such models in terms of tradeoff curves with the goal of high **success rate** on clean examples and low **failure rate** on adversarial examples. Existing untargeted attacks developed for models that do not use confidence thresholding tend to underestimate such models' vulnerability. We propose the `MaxConfidence` family of attacks, which are optimal in a variety of theoretical settings, including one realistic setting: attacks against linear models. Experiments show the attack attains good results in practice. We show that simple defenses are able to perform well on MNIST but not on CIFAR, contributing further to previous calls that MNIST should be retired as a benchmarking dataset for adversarial robustness research. We release code for these evaluations as part of the `cleverhans` (Papernot et al., 2018) library [1].

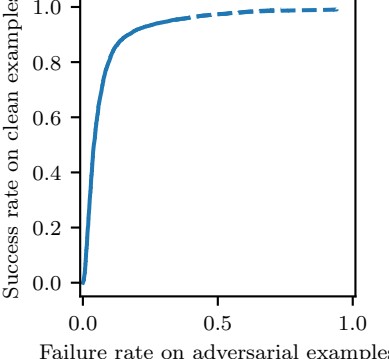 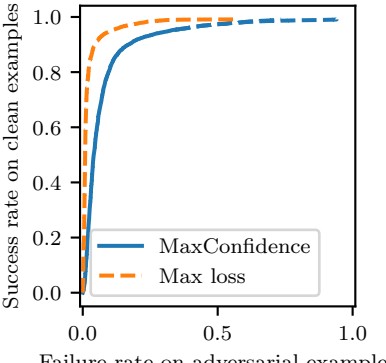

Figure 1:     *(Left)* A success-fail curve. The horizontal axis shows the failure rate (fraction of examples for which the model returns a response and the response is incorrect) on adversarial examples. The vertical axis shows the success rate (fraction of examples for which the model returns a response and the response is correct) on clean examples. Each point on the curve corresponds to using a different confidence threshold. Dashed lines indicate estimates of upper and lower bounds (section 6.1) for thresholds $t < \frac{1}{2}$ where our attack is not provably optimal. The upper and lower bounds are nearly equal so the two bounds are not separately visible at most points. *(Right)* We compare our new `MaxConfidence` attack to a traditional attack that maximizes the loss (Madry et al., 2017). The traditional attack is designed to cause a high loss, not a high failure rate in the presence of confidence thresholding. Our `MaxConfidence` attack performs better or equal at every point on the curve, with the most pronounced differences in the parts of the curve where confidence thresholding is the most effective.

---

[1]ICLR reviewers should be careful not to look at who contributed these features to `cleverhans` to avoid de-anonymizing this submission

# 1 INTRODUCTION

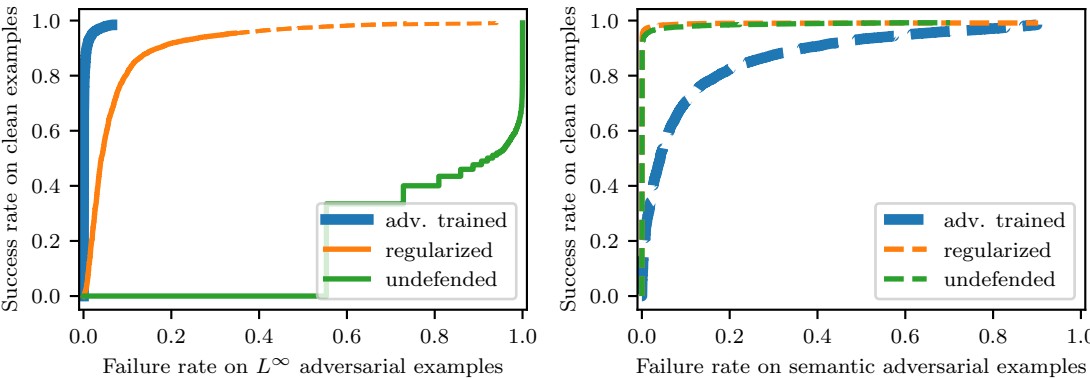

Figure 2: *(Left)* Success-fail curves for three MNIST models on $L^\infty$-constrained adversarial examples. Adversarial training offers a high amount of robustness. The regularized model (appendix B.1) performs well but not as well as adversarial training. Note however, that the regularized model is not specialized for the $L^\infty$ setting. The undefended model performs poorly. The staircase pattern for the undefended model results because few examples lie near the middle of the curve, and when drawing the plot we connect consecutive points in a staircase pattern in order to show the maximum possible failure rate. A curve formed with linear connections between points may false imply that some (success, failure) points are feasible even though they were not reached in the experiment. *(Right)* Success-fail curves for the same MNIST models on semantic adversarial examples. Surprisingly, adversarial training now performs worse than the undefended baseline. The regularized model performs slightly better than the undefended baselines. This illustrates some of the advantages of generic regularization combined with confidence thresholding: models that are generally skeptical of a wide variety of unusual data can avoid being fooled in settings that the designer did not need to explicitly specify ahead of time.

Adversarial examples "are inputs to machine learning models that an attacker has intentionally designed to cause the model to make a mistake" (Goodfellow et al., 2017). All currently known models are highly vulnerable to most known kinds of adversarial examples. See Goodfellow (2018) for one summary of important open problems and research directions.

One popular approach to defending against adversarial examples is to refuse to classify adversarial examples while still providing predictions for clean examples. The evaluation methodology for such defenses has been of limited quality. In this work, we present both metrics and attack methods for security evaluation of models that use confidence thresholding to refuse to classify some examples.

Models based on confidence thresholding have the potential to solve many open problems in adversarial robustness:

- The current state of the art in adversarial robustness (Madry et al. (2017), and with some caveats, Kannan et al. (2018)) is based on adversarial training (Szegedy et al., 2013; Goodfellow et al., 2014b; Warde-Farley & Goodfellow, 2016; Kurakin et al., 2016). So far adversarial training tends to result in robustness to the attack used during training but does not result in robustness to other kinds of attacks (e.g. Sharma & Chen (2017) showed that adversarially trained models designed to be robust to small perturbations as measured by the $L^\infty$ norm do not also become robust to small perturbations as measured by the $L^1$ norm). It seems unlikely that adversarial training will scale to make the model give the right answer everywhere in the model's input domain, but maybe models based on confidence thresholding can learn to assign low confidence almost everywhere in the input domain, and thereby resist a very wide range of potential attacks. Confidence thresholding thus seems like a viable candidate to solve harder problems involving adversarial examples with fewer restrictions (Brown et al., 2018).

- Adversarial training has high computational cost, but maybe models with appropriate confidence can be trained at lower cost. We show that this is feasible on MNIST, and our evaluation methods pave the way to test future models on more difficult datasets.

- Confidence thresholding offers a simple, flexible way to trade performance on clean data for robustness to adversarial examples, simply by adjusting the threshold $t$. Our proposed evaluation methods can automatically produce entire tradeoff curves after running our proposed attack only once per example. Choosing the right point on the tradeoff curve does not require retraining the model or re-running the evaluation for each hypothetical threshold.

- Adversarial training often causes a reduction in accuracy on naturally occurring data (Su et al., 2018). Models that use confidence thresholding can actually achieve higher accuracy than undefended models, albeit with reduced coverage. When a model reduces coverage, it reduces the total volume of input space for which it can have an error. Other methods run the risk of moving errors into locations not measured by the current benchmark.

Many proposed defenses have been based on detecting adversarial examples and then refusing to classify them. Many of these proposed defenses have been broken (Carlini & Wagner, 2017). We observe that confidence thresholding is different from detection in a subtle but important way. In the case of confidence thresholding, refusing to return a response does not imply that the input is necessarily an "adversarial example". The classifier is free to shut off for a wide variety of inputs, so long as they do not appear often in the naturally occuring data. This makes the confidence thresholding defense much less likely to overfit to a specific conception of what an adversarial example looks like.

Despite this potential, it has not yet been clear how to evaluate models based on confidence thresholding. In this work, we advocate analyzing models in terms of a tradeoff curve: models should have high **success rate** on naturally occurring data and low **failure rate** on adversarial data, defined in section 3. Models that use confidence thresholding must also be benchmarked appropriately, against attacks that are aware the defender will use confidence thresholding. We propose a new family of attacks called `MaxConfidence` that is optimal in a variety of theoretical settings, and optimal in practice against linear models.

## 2 ACCURACY AND COVERAGE

**Accuracy** and **coverage** are metrics that are used to evaluate machine learning models that are capable of deciding whether or not to provide an answer for each input example (Kohavi & Provost, 1998). If a model chooses to provide an answer for an example, that example is called **covered**. *Coverage* is the fraction of examples that are covered by the model, while *accuracy* is the fraction of covered examples that are correctly classified. There is a tradeoff between accuracy and coverage: most machine learning papers evaluate accuracy at 100% coverage, and as we reduce coverage accuracy in the remaining examples should generally increase. At the extreme of zero coverage, we regard a fraction of $\frac{0}{0}$ as 100% accuracy.

Many different mechanisms can be used to decide whether or not to cover an example. In this work, we focus on models that use confidence thresholding. These models estimate the probability of each class, and cover an example only if the probability of the most likely class exceeds some threshold. Specifically, the confidence of a model is given by $c(\boldsymbol{x}) = \arg\max_y p_{\text{model}}(y \mid \boldsymbol{x})$ and $\boldsymbol{x}$ is covered if $c(\boldsymbol{x}) > t$. This threshold can be chosen to optimize tradeoffs between accuracy and coverage[2]. As we will see, our `MaxConfidence` attack is optimal under some theoretical idealizations for $t \geq \frac{1}{2}$. This corresponds to classifiers that return a response only if they believe that they are more likely to be correct than incorrect. In some contexts, it may be useful to have a classifier make its best guess, even if it expects that guess to be wrong ($t < \frac{1}{2}$). In this case, we can bound (under some theoretical idealizations) how much `MaxConfidence` underestimates the true error rate, and we find experimentally that the underestimate is small.

---

[2]These and other definitions are repeated for the reader's convenience a notation section: appendix A.

## 3 SUCCESS RATE AND FAILURE RATE

Traditionally, confidence thresholding models have been evaluated in terms of how they trade accuracy for coverage on a single test distribution, usually with the goal of avoiding having ordinary test errors (see e.g. Goodfellow et al. (2014a)). In this work we advocate a new evaluation methodology, intended for adversarial situations. We propose to measure the **success rate** (fraction of examples that are both covered and correct) on the naturally occurring test set and the **failure rate** (fraction of examples that are both covered and incorrect) on adversarial inputs. Intuitively, success rate measures how well the model can be used for its intended purposes and failure rate measures how effectively an adversary can control the model. Each model has not just a single success rate and failure rate. Different values of the confidence threshold $t$ yield a tradeoff curve. Extremely high values of $t$ result in zero success and zero failure, while lower values of $t$ cover more examples and open the opportunity for both success and failure. Good models will have a success rate that grows faster than the failure rate as $t$ shrinks. Different applications will have different requirements in terms of whether it is better to seek success or avoid failure, and the designer can easily use these curves to choose the appropriate threshold. See figure 1 for an example of a success-failure curve, using our Model A (appendix B) on the MNIST dataset. The curve is generated by running the model once on clean data and once on adversarial data (presuming the adversarial data is generated using our `MaxConfidence` attack, which is optimal or at least produces a bound for all thresholds), recording the probability assigned to each output example. The whole curve can then be swept out cheaply by sorting the probabilities and plotting the success rate and failure rate that would be obtained by every possible threshold. There is no need to retrain the model or rerun the attack for each considered threshold.

Defenses based on incomplete coverage are appropriate for some applications but not others. Such defenses remain vulnerable to denial of service attacks (the attacker repeatedly sends many examples that are not covered). Such defenses are not appropriate for applications where continuous performance is important for security For example, imagine an autonomous delivery drone that relies on a machine learning system to continuously output flight control commands. If the drone is defended using a system based on reducing coverage, an adversary could crash the drone by exposing it to adversarial examples that cause its flight control system to refuse to produce output. In other applications, incomplete coverage can be useful. For example, consider adversarial attacks against speech recognition systems (Carlini et al., 2016). In this setting it is perfectly acceptable to respond to adversarial examples with something along the lines of "I didn't understand that" because the user does not depend on the phone responding to input provided by the adversary.

Success-fail curves do not need to be used only with worst-case norm-constrained perturbations as is common in the adversarial example literature. When measuring the failure rate on adversarial data, it is possible in principle to evaluate based on worst-case inputs drawn from sets of input space that are more complicated than norm balls, or to evaluate based on randomly sampled inputs from a distribution that is more difficult than the clean data. For example, one may want to evaluate a neural network on CAPTCHAs (Goodfellow et al., 2014a). Indeed, we show a success-failure plot for a random attack not based on norm balls in section 7 and section 8.

## 4 LIMITATIONS OF PREVIOUS ATTACKS

We introduce the `MaxConfidence` attack to address two shortcomings of previous attacks. First, previous attacks were often not designed to adapt to overcome confidence thresholding defenses. Figure 1(right) shows that traditional untargeted attacks do not maximally exploit confidence thresholding models—it is necessary to switch to the `MaxConfidence` attack. Second, we are not aware of any attack against confidence thresholding models that offers an optimality guarantee, even in unrealistic theoretical settings. Our proposed attack is optimal against linear models, and is optimal against general models under the assumption that the underlying optimization algorithm solves the optimization problem with sufficient accuracy (section 6.2).

### 4.1 OPTIMALITY AGAINST LINEAR MODELS

Goodfellow et al. (2014b) showed that the **fast gradient sign method** attack causes the most cross-entropy loss possible against logistic regression under an $L^\infty$ constraint on the adversarial example.

This optimality guarantee does not generalize from logistic regression (2-class linear classifier) to softmax regression ($k$-class linear classifier).

Given a clean input $\boldsymbol{x}$, an adversarial example construction algorithm must find an adversarial example $\tilde{\boldsymbol{x}} \in \mathbb{S}(\boldsymbol{x})$, a set of inputs that should be equivalent to $\boldsymbol{x}$. Provided that the attack set $\mathbb{S}(\boldsymbol{x})$ is convex, the `MaxConfidence` attack solves a different convex optimization problem for every step of the attack. When the attack is implemented with a proper convex optimization algorithm that can provide a certificate of optimality at convergence, the `MaxConfidence` attack is thus optimal for $k$ class linear models, provided that $t \geq \frac{1}{2}$.

By comparison, previous untargeted attacks were not optimal. Previous untargeted attacks aim to *maximize* the negative log probability the linear model assigns to the true class. Minimizing such a loss is a convex optimization problem, but maximizing it is not. For $k > 2$, there can be multiple non-equivalent local maxima. (We speculate that this is the reason that adversarial training performs much better when initialized with a noisy starting point)

### 4.2 TARGETED ATTACKS

A variety of targeted attacks for maximizing $p_{\text{model}}(y \mid \tilde{\boldsymbol{x}})$ for a specific class $y$ already exist (Szegedy et al., 2013; Goodfellow et al., 2014b; Kurakin et al., 2016). These attacks are already optimal in the same idealized settings where `MaxConfidence` is optimal, even against confidence thresholding models, in terms of *the rate of hitting the specific target $y$*. However, pre-existing targeted attacks do not on their own find the maximal *failure* rate. Adversarial machine learning is often a non-zero-sum game, in which the defender's cost increases with every mistake but the attacker's cost decreases only with a hit on a specific target class. The `MaxConfidence` attack is thus useful for evaluations that are more relevant to designing a well-defended system.

## 5 THE `MaxConfidence` ATTACK

The `MaxConfidence` family of attacks is simple. Given a naturally occuring input $\boldsymbol{x}$, a true class label $y^*$, a set of allowed attack points $\mathbb{S}$, and an optimization algorithm $\mathcal{O}$ such that $\mathcal{O}(f, \mathbb{S}) \approx \arg\max_{\boldsymbol{x} \in \mathbb{S}} f(\boldsymbol{x})$, the `MaxConfidence` attack uses the following procedure:

1. Let $\boldsymbol{x}^{(i)} = \mathcal{O}(p(y = i \mid \boldsymbol{x}), \mathbb{S})$ for all values of $i$ from 1 to $k$ except for $y^*$. In other words, use the chosen optimization algorithm to run a targeted attack on each of the wrong classes.

2. Return $\boldsymbol{x}^{(i*)}$ for $i^* = \arg\max_{i \neq y^*} \max_{j \neq y^*} p(y = j \mid \boldsymbol{x}^{(i)})$. In other words, out of all the targeted attacks, choose the one that resulted in the most confidence on a wrong class. (For a perfect optimization algorithm, the optimal $j$ will always be $i$, but imperfect optimization algorithms may sometimes inadvertently return points where the highest probability is assigned to a class that was not intentionally targeted)

When working in settings where no actual optimization algorithm $\mathcal{O}$ is guaranteed to be optimal, the performance of the attack can be improved by **attack bundling**: rather than taking the max across only different target classes, additionally take the max across runs of many different algorithms $\mathcal{O}$ or multiple calls to stochastic optimization algorithms. This can include optimization algorithms based on random search, black box transfer, etc. Examples of reasonable choices for $\mathcal{O}$ include the optimization algorithms used in existing targeted attacks (section 4.2). All gradient-based optimizers must be implemented with care for numerical stability: the objective should be implemented in log space, using a numerically stable implementation of the log-softmax function, etc.

## 6 PROOF OF OPTIMALITY OF THE IDEALIZED `MaxConfidence` ATTACK

Consider a theoretical idealization of the `MaxConfidence` attack, under the assumption that the optimization algorithm is perfect. This is followed by a discussion of how the optimization algorithm need only approximately maximize the confidence (section 6.2).

Given a clean input $\boldsymbol{x}$ with correct label $y$, the theoretical idealization of the `MaxConfidence` attack finds an adversarial input $\tilde{\boldsymbol{x}}$ such that

$$\tilde{\boldsymbol{x}} = \arg\max_{\boldsymbol{x} \in \mathbb{S}(\boldsymbol{x})} \left[ \max_{y \neq y^*} p_{\text{model}}(y \mid \tilde{\boldsymbol{x}}) \right]. \tag{1}$$

We now show that, for every clean example $(\boldsymbol{x}, y^*)$, if any attack restricted to providing inputs from $\mathbb{S}(\boldsymbol{x})$ can find a covered and misclassified example for a classifier that works by confidence thresholding $p_{\text{model}}$ with threshold $t \geq \frac{1}{2}$, the `MaxConfidence` attack will also find a covered and misclassified example. This implies that the MaxConfidence attack obtains the optimal failure rate when $\boldsymbol{x}$ is sampled from some distribution.

The proof proceeds by cases. In the first case, suppose that for some input $\boldsymbol{x}$, `MaxConfidence` finds an input $\tilde{\boldsymbol{x}}$ that is covered and misclassified. Then there is nothing further to show for this case.

In the second case, suppose that `MaxConfidence` finds an input $\tilde{\boldsymbol{x}}$ that is not covered. Suppose there is some other input $\hat{\boldsymbol{x}} \in \mathbb{S}(\boldsymbol{x})$ that is covered. We thus have $\max_y p(y \mid \hat{\boldsymbol{x}}) > t$ while $\max_y p(y \mid \tilde{\boldsymbol{x}}) < t$. This implies that $\arg\max_y p(y \mid \hat{\boldsymbol{x}}) = y^*$, otherwise the `MaxConfidence` attack would have selected $\hat{x}$ instead of $\tilde{x}$. Thus $\hat{x}$ is correctly classified by the model and does not contribute to the failure rate.

In the third and final case, suppose that `MaxConfidence` finds an input $\tilde{\boldsymbol{x}}$ that is covered but is correctly classified. Recall that we have required that the confidence threshold $t \geq \frac{1}{2}$. Also, recall that in our definition an example is covered only if its confidence *strictly exceeds* the threshold. Because we have assumed that $\tilde{\boldsymbol{x}}$ is covered and correctly classified, we know $p_{\text{model}}(y = y^* \mid \tilde{\boldsymbol{x}}) > t$. Now suppose that there exists an example $\hat{\boldsymbol{x}} \in \mathbb{S}(\boldsymbol{x})$ that is covered and is incorrectly classified as belonging to some false class, $f$, such that $f \neq y^*$. Thus

$$p_{\text{model}}(y = f \mid \hat{\boldsymbol{x}}) > t. \tag{2}$$

The `MaxConfidence` attack chose $\tilde{\boldsymbol{x}}$, so there must exist a false class $g$, such that $g \neq y^*$ and

$$p_{\text{model}}(y = g \mid \tilde{\boldsymbol{x}}) \geq p_{\text{model}}(y = f \mid \hat{\boldsymbol{x}}). \tag{3}$$

Because a probability distribution must sum to 1: $\quad p_{\text{model}}(y = y^* \mid \tilde{\boldsymbol{x}}) + p_{\text{model}}(y = g \mid \tilde{\boldsymbol{x}}) \leq 1.$

From equation 3, we have: $\quad p_{\text{model}}(y = y^* \mid \tilde{\boldsymbol{x}}) + p_{\text{model}}(y = f \mid \hat{\boldsymbol{x}}) \leq 1.$

From equation 2, we have: $\quad p_{\text{model}}(y = y^* \mid \tilde{\boldsymbol{x}}) + t \leq 1.$

Algebraic rearrangement gives: $\quad p_{\text{model}}(y = y^* \mid \tilde{\boldsymbol{x}}) \leq 1 - t.$

When $t \geq \frac{1}{2}$, this results in a contradiction: we have defined $p_{\text{model}}(y = y^* \mid \tilde{\boldsymbol{x}})$ to be strictly greater than the threshold but then shown it is less than or equal to the threshold.

## 6.1 Bounding failure rate for $t < \frac{1}{2}$

The attack may not be optimal when we relax the requirement that the confidence threshold $t \geq \frac{1}{2}$. Consider the counterexample where:

$$p_{\text{model}}(y = y^* \mid \tilde{\boldsymbol{x}}) = .8 \quad p_{\text{model}}(y = g \mid \tilde{\boldsymbol{x}}) = .2$$
$$p_{\text{model}}(y = y^* \mid \hat{\boldsymbol{x}}) = .1 \quad p_{\text{model}}(y = f \mid \hat{\boldsymbol{x}}) = .11$$

Conceptually, the counterexample shows that `MaxConfidence` fails because there is a corner case where the model misclassifies the input without concentrating probability mass on any single wrong class. Adding the requirement that $t \geq \frac{1}{2}$ requires the model to concentrate probability on a class for the example to be covered.

If one wishes to study a model with threshold $t < \frac{1}{2}$, the `MaxConfidence` attack is still a useful tool. The attack is no longer guaranteed to be optimal but can be used to bound the failure rate under optimal attack. We know that the optimal failure rate must be less than or equal to the coverage of the `MaxConfidence` attack. This is because case 3 above (examples returned by `MaxConfidence` that are covered and correctly classified) indicates an opportunity for a stronger attack to find an attack that is both covered and misclassified. The other two cases do not present opportunities for a stronger attack to increase the failure rate. The optimal failure rate is thus lower bounded by the failure rate of `MaxConfidence` and upper bounded by the coverage of the `MaxConfidence`. (Keep in mind that all of this is in theory, assuming the optimization algorithm yields a sufficiently good approximation. In reality, attacks can provide lower bounds but not upper bounds on failure rate, due to imperfect optimization)

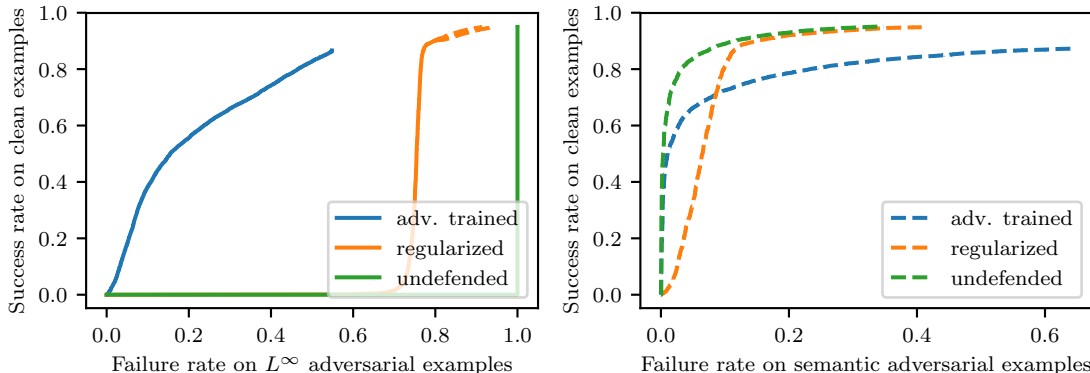

Figure 3: *(Left)* Success-fail curves for three CIFAR-10 models on $L^\infty$-constrained adversarial examples. Adversarial training offers a high amount of robustness. Unlike on MNIST, the regularized model does not perform well. *(Right)* Success-fail curves for the same CIFAR-10 models on semantic adversarial examples. Like on MNIST, the baseline, with no defense other than confidence thresholding, performs best. Unlike on MNIST, here the regularized and adversarially trained models are somewhat comparable. Each model is better for one part of the tradeoff curve. Indeed, their curves intersect at approximately (success=70%, failure=9%). For applications where achieving success is more important, it is preferable to use the regularized model. For applications where avoiding failure is more important, it is preferable to use the adversarially trained model.

## 6.2 APPROXIMATE OPTIMIZATION

So far we have discussed the optimality proof assuming a perfect optimization algorithm. In fact, the proof requires only that if there exists a misclassified any input $\hat{x} \in \mathbb{S}(x)$ with confidence $c(\hat{x}) > t$, then the optimization algorithm $\mathcal{O}$ will find a misclassified input $\tilde{x} \in \mathbb{S}(x)$ that also has confidence $c(\tilde{x}) > t$. This leaves considerable room for the optimization algorithm to underestimate the true maximum confidence. In the most extreme case, where $c(\hat{x}) = 1$ and $c(\tilde{x})$ approaches $\frac{1}{2}$ from above, the optimization algorithm may safely underestimate by an amount approaching $\frac{1}{2}$ from below. Despite this underestimation, the optimization algorithm still obtains the optimal failure rate. In practice, however, we expect that most optimization algorithms will suffer from extreme underestimation periodically, so our estimates of the failure rate should be regarded as lower bounds. In the future, it may be possible to make success-fail curves that upper bound the failure rate using verification methods.

## 7 MNIST EXPERIMENTS

On MNIST, we studied three different models on two different kinds of adversarial examples. We compared the adversarially trained model of Madry et al. (2017) to two versions of our "Model A" (appendix B). Model A is a simple convolutional network. We find that when strongly regularized (appendix B) Model A obtains very different success-fail curves than the undefended baseline, even though the two have relatively similar accuracy at 100% coverage for both clean examples and adversarial examples. This suggests that success-fail curves are an interesting metric that reveal otherwise hidden benefits of models that have good confidence estimates. We find that the regularized Model A actually performs reasonably well against `MaxConfidence` using 40 steps of projected gradient descent starting from a random initial starting point as the inner optimizer and a limit on the $L^\infty$ norm of $\epsilon = 0.3$. While the regularized Model A does not perform nearly as well as adversarial training in this threat model, recall that adversarial training costs roughly 40X more (due to the need to generate adversarial examples during the inner loop of training) and that adversarial training is specialized for this exact threat model. We also evaluate all three models on "semantic adversarial examples" (Hosseini et al., 2017) formed by taking the negative of each image. These negative images can still be recognized accurately by humans but are difficult for modern convolutional networks. In the context of this study, our main goal is for the model to obtain low confidence on the

unusual, out of domain negative images while maintaining high confidence on the clean data. Surprisingly, the adversarially trained model actually performs *worse than the undefended baseline* on this task. Of course, the adversarially trained model was not designed for this task, nor was it advertised as being suitable for this task. Nonetheless, this illustrates a limitation of adversarial training: the designer must specify the threat model exactly. Methods based on confidence thresholding are able to have generically low confidence on inputs that differ from the training distribution, and do not require the designer to enumerate all contingencies. Results are shown in figure 2.

## 8    CIFAR-10 EXPERIMENTS

We repeated our experiments on CIFAR and found that confidence thresholding applied to a regularized model (appendix C is not nearly as effective at resisting $L^\infty$ adversarial examples on this dataset. On MNIST, we found that the regularized model outperformed the adversarially trained model at classifying semantic adversarial examples. On CIFAR-10, we now find that the adversarially trained model is better at one end of the curve while the regularized model is better at the other end of the curve. One finding did hold up from MNIST: we still find that the adversarially trained model is worse than a baseline that uses no defense other than confidence thresholding on semantic adversarial examples. The models used in both cases are those from Madry et al. (2017). For the adversarially trained model, we used their pretrained checkpoint. For the undefended model, we used their code, with the adversarial example generation commented out, so that the same model was trained on clean rather than adversarial examples. For the regularized model, we retrained their model, substituting in our own regularization (the same as the regularization described in appendix B). Results are shown in figure 3.

Our findings are consistent with those of Carlini & Wagner (2017): robustness properties on MNIST do not transfer well to CIFAR-10. We thus strongly endorse their recommendation to "evaluate on more than MNIST". We recommend retiring MNIST as a benchmark for adversarial examples because too many simple techniques show significant robustness on this dataset but not others.

## 9    CONCLUSION

We have made the following contributions:

- We have shown that adversarial training on one kind of out-of-distribution data can actually worsen performance on other kinds of out-of-distribution data, relative to a baseline that uses confidence thresholding as the only defense.
- We have introduced the evaluation methodology of success-fail curves, showing which success rates on clean data and failure rates on adversarial data are feasible for different confidence thresholds.
- We have presented an attack that is optimal against a variety of confidence thresholding models. Specifically, it is optimal against linear classifiers and optimal against general models whenever the underlying optimization approximately succeeds.
- We have shown an evaluation methodology that maps out an entire success-failure tradeoff curve without needing to re-train the model or re-run the evaluation for different thresholds.
- We have shown that confidence thresholding with simple regularization is sufficient to achieve reasonable robustness to $L^\infty$ attacks on MNIST, despite being roughly 40X cheaper to train than adversarial training.
- We have shown that confidence thresholding can lead to robustness to a variety of attacks, without needing to anticipate and formally specify each attack type.

Overall, we hope that our evaluation methodology will help to design and rigorously test low-cost, versatile defenses against a wide variety of adversarial examples.

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

## A  NOTATION AND DEFINITIONS

| | |
|---|---|
| $\boldsymbol{x}$ | The input to the model |
| $y$ | A random variable representing the class label for an example |
| $y^*$ | The true label for $\boldsymbol{x}$, i.e. the correct value of $y$ |
| $\mathbb{S}(\boldsymbol{x})$ | A set of points that should be equivalent to $\boldsymbol{x}$. An adversarial example corresponding to $\boldsymbol{x}$ must be drawn from this set. |
| $\tilde{\boldsymbol{x}}$ | An adversarial example corresponding to $\boldsymbol{x}$ |
| $p_{\text{model}}(y \mid \boldsymbol{x})$ | The conditional distribution over the classes represented by the model |
| $c(\boldsymbol{x})$ | $\arg\max_y p_{\text{model}}(y \mid \boldsymbol{x})$, the confidence of the model for input $\boldsymbol{x}$ |
| $k$ | The number of classes |
| $t$ | The confidence threshold used by the model. An input $\boldsymbol{x}$ is covered only if $c(\boldsymbol{x}) > t$. |
| $\boldsymbol{w}$ | The weight vector for a logistic regression model |
| $\boldsymbol{\eta}$ | A perturbation applied to a clean input $\boldsymbol{x}$ |

## B  MODEL A

Our "Model A" is a simple model that we tuned by trial and error to yield better success-fail curves than the baseline. We do not advocate "Model A" as the latest and greatest model that everyone should switch to. It is only included as a test point to show that our evaluation methodology can find interesting differences between models that have similar accuracy at 100% coverage on clean and adversarial data.

Our trial-and-error design process was based on performance on clean data and on $L^\infty$ adversarial examples. We did not use information about performance on semantic adversarial examples during the design process, so the defense was not designed in any specific way to handle these examples.

The model architecture is straightforward to describe in `cleverhans` format:

```
layers = [
    Conv2D(nb_filters, (3, 3), (2, 2), "SAME"),
    ReLU(),
    Add([Conv2D(nb_filters, (3, 3), (1, 1), "SAME"),
        ReLU(),
        Conv2D(nb_filters, (3, 3), (1, 1), "SAME")]),
    Conv2D(nb_filters * 2, (3, 3), (2, 2), "SAME"),
    ReLU(),
    Conv2D(nb_filters * 2, (3, 3), (1, 1), "VALID"),
    ReLU(),
    Flatten(),
    Linear(nb_classes),
    Softmax()]
```

In other words, it is a simple convolutional network, containing a convolution and 2X downsampling layer, a residual layer, two convolutional layers, and a fully connected layer to output the logits. There are no normalization layers, etc., and all of the hidden units are ReLUs (Jarrett et al., 2009; Nair & Hinton, 2010; Glorot et al., 2011).

### B.1  REGULARIZATION

In some experiments we regularized Model A to have better confidence estimates. Previously, we have experimented with traditional techniques for improving confidence estimates, such as weight decay and approximate Bayesian inference, and found that they are ineffective in the adversarial setting. We present here regularization methods that we found by trial and error to be somewhat

effective in the adversarial setting. We emphasize that we are not advocating these regularization methods as the latest and greatest method that everyone should use—we offer them as an example to show that our evaluation methods can find interesting differences between models that have similar accuracy at full coverage. In the future, we hope that others will find methods that are both more effective and more principled.

Through trial and error, we found that three techniques greatly improved the confidence estimates of our model in the adversarial setting: **extreme label smoothing**, **variable label smoothing**, and **noisy logit pairing**.

### B.1.1  EXTREME LABEL SMOOTHING

Label smoothing (Szegedy et al., 2015) is a regularization technique in which the hard one-hot labels for a classifier are replaced with soft labels, where some amount of probability $\delta$ is removed from the correct class and distributed among the other classes. For example, an example in class 5 for a 10-class problem has one-hot label

$$[0, 0, 0, 0, 0, 1, 0, 0, 0, 0].$$

With $\delta = .1$, this would be smoothed to

$$[.1/9, .1/9, .1/9, .1/9, .1/9, .9, .1/9, .1/9, .1/9, .1/9].$$

Label smoothing has previously been observed to cause a moderate amount of robustness to weak adversarial examples (Warde-Farley & Goodfellow, 2016).

Label smoothing is typically used with small values of $\delta$ and can be thought of as modeling a small probability that each example is unlabeled.

We found that we obtain much better success-fail curves using very large values of $\delta$, where $1 - \delta$ approaches $\frac{1}{2}$.

### B.1.2  VARIABLE LABEL SMOOTHING

We found that we obtained best results when we used different label smoothing for different examples. Specifically, we trained on both clean and noisy examples, with the noisy examples formed by adding unit Gaussian noise to the clean examples (yes, unit Gaussian is quite large). We used a larger value of $\delta$ for the label smoothing on the noisy examples than on the clean examples. Our motivation is that this should encourage the model to have local maxima of confidence on clean examples, with confidence diminishing in all directions radiating out from clean examples. Specifically, we used $\delta = 0.3$ for clean examples and $\delta = 0.4$ for noisy examples.

### B.1.3  NOISY LOGIT PAIRING

We found that we obtained better success-fail curves when we additionally penalized the mean squared error between the logits on clean and noisy examples. This is similar to adversarial logit pairing (Kannan et al., 2018), but using noisy examples rather than adversarial examples.

Noisy logit pairing encourages the model to learn smoothing functions. When applied to logistic regression, it is equivalent to weight decay. For a logistic regression model with weights $\boldsymbol{w}$ and noisy perturbation $\boldsymbol{\eta}$ we have:

$$(\boldsymbol{w}^\top \boldsymbol{x} - \boldsymbol{w}^\top (\boldsymbol{x} + \boldsymbol{\eta}))^\top (\boldsymbol{w}^\top \boldsymbol{x} - \boldsymbol{w}^\top (\boldsymbol{x} + \boldsymbol{\eta}))$$
$$= (\boldsymbol{w}^\top \boldsymbol{\eta})^\top (\boldsymbol{w}^\top \boldsymbol{\eta})$$
$$= (\boldsymbol{w}^T \boldsymbol{\eta})^2.$$

For any noise distribution where $\boldsymbol{\eta}$ has identity covariance, this simplifies to $\boldsymbol{w}^\top \boldsymbol{w}$. Thus for logistic regression, noisy logit pairing is equivalent to weight decay.

We used a coefficient of 1 on our logit pairing loss.

## C   REGULARIZED CIFAR-10 MODEL

Our regularized CIFAR-10 model was created by modifying the code for Madry et al. (2017) to use the loss function described in appendix B, with the following modifications:

- Model A did not use weight decay. We included weight decay, simply because it is part of the Madry et al. (2017) codebase and we wanted to make as few changes as possible to make comparison straightforward.

- On MNIST, we trained Model A on noisy examples created using very large, unit Gaussian perturbations, for data in the range [-1, 1]. On CIFAR-10, we found that such large perturbations were harmful. Instead we used a standard deviation of $8$ for data in the range [0, 255]. Also, we clipped the noisy examples into the range [0, 255].

- We used $\delta = \frac{1}{4}$ for label smoothing on clean examples (less extreme than we used on MNIST) and $\delta = \frac{3}{4}$ for label smoothing on noisy examples (much more extreme than we used on MNIST). These settings were coarsely tuned for good performance on clean data and $L^\infty$ adversarial examples but were not tuned for performance on semantic adversarial examples. On MNIST, we did not tune the logit smoothing coefficient at all, and just used a value of $1$. On CIFAR-10, we used $1.2$.

