# OpenReview forum: "Evaluation Methodology for Attacks Against Confidence Thresholding Models"
_ICLR.cc/2019/Conference_

### Official Review · AnonReviewer2 · 2018-10-31
**Interesting attack but an unclear paper with limited experimental support.**

**Rating:** 4
**Confidence:** 4

**Review:**

The paper presents an evaluation methodology for evaluating attacks on confidence thresholding methods and proposes a new kind of attack. In general I find the writing poor, as it is not exactly clear what the focus of the paper is - the evaluation or the new attack? The experiments lacking and the proposed evaluation methodology & theoretical guarantees trivial.

Major remarks:
- Linking the code and asking the reviewers not to look seems like bad practice and close to violating double blind, especially when considering that the cleavhans library is well known. Should have just removed the link and cleavhans name and state it will be released after review.

- It is unclear what the focus of the paper is, is it the evaluation methodology or the new attack? While the evaluation methodology is presented as the main topic in title, abstract and introduction most of the paper is dedicated to the attack.

- The evaluation methodology is a good idea but is quiet trivial. Also, curves are nice visually but hard to compare between close competitors. A numeric value like area-under-the-curve should be better.

- The theoretical guarantees is also quiet trivial, more or less saying that if a confident adversarial attack exists then finding the most confident attack will be successful. Besides that the third part of the proof can be simplified significantly.

- The experiments are very lacking. The authors do not compare to any other attack so there is no way to evaluate the significance of their proposed method

- That being said, the max-confidence attack by itself sounds interesting, and might be useful even outside confidence thresholding.

- One interesting base-line experiment could be trying this attack on re-calibrated networks e.g. “On Calibration of Modern Neural Networks”

- Another baseline for comparison could be doing just a targeted attack with highest probability wrong class.

- I found part 4.2 unclear

- In the conclusion, the first and last claims are not supported by the text in my mind.



Minor remarks:

- The abstract isn’t clear jumping from one topic to the next in the middle without any connection.

- Having Fig.1 and 2 right on the start is a bit annoying, would be better to put in the relevant spot and after the terms have been introduced.

-In 6.2 the periodically in third line from the end seems out of place.

---

### Official Review · AnonReviewer3 · 2018-11-02
**Hard to understand**

**Rating:** 3
**Confidence:** 3

**Review:**

This paper proposes an evaluation method for confidence thresholding defense models, as well as a new approach for generating of adversarial examples by choosing the wrong class with the most confidence when employing targeted attacks.

Although the idea behind this paper is fairly simple, the paper is very difficult to understand.  I have no idea that what is the propose of defining a new evaluation method and how this new evaluation method helps in the further design of the MaxConfidence method. Furthermore, the usage of the evaluation method unclear as well, it seems to be designed for evaluating the effectiveness of different adversarial attacks in Figure 2. However, in Figure 2, it is used for evaluating defense schemes. Again, this confuses me on what is the main topic of this paper. Indeed, why the commonly used attack success ratio or other similar measures cannot be used in the case? Intuitively, it should provide similar results to the success-failure curve.

The paper also lacks experimental results, and the main conclusion from these results seems to be "MNIST is not suitable for benchmarking of adversarial attacks". If the authors claim that the proposed MaxConfidence attack method is more powerful than the MaxLoss based attacks, they should provide more comparisons between these methods.

Meanwhile, the computational cost on large dataset such as ImageNet could be huge, the authors should further develop the method to make sure it works in all situations.

---

> ### Author Response · Authors · 2018-11-06
> **Reply**
>
> The main topic of the paper is how to evaluate models that use confidence thresholding. The primary purpose is to compare *defenses*. However, to justify the attack strategy that we propose to use, we also compare *attacks*. Specifically, we provide an experiment demonstrating that our attack actually is stronger than the baseline. However, it is not really necessary to provide multiple experiments demonstrating that MaxConfidence is more powerful because the superiority of MaxConfidence is theoretically guaranteed.

---

### Official Review · AnonReviewer1 · 2018-11-08
**A good topic to explore, but suffers from methodological problems**

**Rating:** 2
**Confidence:** 4

**Review:**

This paper introduces a family of attack on confidence thresholding algortihms. Such algorithms are allowed to refuse to make predictions when their confidence is below a certain threshold.

There are certainly interesting links between such models and KWIK [1] algorithms (which are also supposed to be able to respond 'null' to queries), however they are not mentioned in this paper, which focuses mainly on evaluation methodologies.

The definition of the metric is certainly natural: you would expect some trade-off between performance in the normal versus the adversarial regime. I am not certain why the authors don't simply measure the success rate on both natural and adversarial conditions, so as to have the performance metric uniform. Unfortunately the paper's notationleaves something to be desired, as it fails to concretely define the metric.
Let me do so instead, and consider the classification accuracy of a classification rule $P_t$ using a threshold $t$ under a (possibly adaptive) distribution $Q$ to be $U(P,Q)$. Then, we can consider $Q_N, Q_A$ as the normal and adversarial distribution and measure the corresponding accuracies.

Even if we do this, however, the authors do not clarify how they propose to select the classification rule. Should they employ something like a convex combination:
\[
V(P_t) := \alpha U(P_t, Q_N) + (1 - \alpha) U(P_t, Q_A)
\]
or maybe take a nimimax approach
\[
V(P_t) := \min \{U(P_t, Q) | Q = Q_A, Q_N\}
\]

In addition, the authors simply plot curves for various choices of $t$, however it is necessary to take into account the fact that measuring performance in this way and selecting $t$ aftewards amounts to a hyperparameter selection [2]. Thus, the thresholding should be chosen on an independent validation set in order to optimise the chosen performance measure, and then the choice should evaluated on a new test set with respect to the same measure $V$

The MaxConfidence attack is not very well described, in my opinion. However, it seems it simply wishes to find to find a single point $x \in \mathbb{S}$ that maximises the probability of misclassification. It is not clear to me why performance against an attack of this type is interesting to measure.

The main contribution of the paper seems to be the generalisation of the attack by Goodfellow et al to softmax regression. The proof of this statement is in a rather obscure place in the paper.

I am not sure I follow the idea for the proof, or what they are trying to prove. The authors should follow a standard Theorem/Proof organisation, clearing stating assumptions and what the theorem is showing us. It seems that they want to prove that if a solution to (1) exists, then MaxConfidence() finds it. But the only definition of MaxConfidence is (1). Hence I think that their theorem is vacuous. There are quite a few details that are also unclear such as what the authors mean by 'clean example' etc.

However the authors do not explain their attack very well, their definition of the performance metric is not sufficiently formal, and their evaluation methodology is weak. Since evaluation methodology is the central point of the paper, this is a serious weaknes. Finally, there doesn't seem to be a lot of connection with the conference's topic.

[1] Li, Lihong, Michael L. Littman, and Thomas J. Walsh. "Knows what it knows: a framework for self-aware learning." Proceedings of the 25th international conference on Machine learning. ACM, 2008.

[2] Bengio, Samy, Johnny Mariéthoz, and Mikaela Keller. "The expected performance curve." International Conference on Machine Learning, ICML, Workshop on ROC Analysis in Machine Learning. No. EPFL-CONF-83266. 2005.

---

### Meta-Review · Area_Chair1 · 2018-12-17
**Reject**

**Confidence:** 5
**Recommendation:** Reject

**Metareview:**

The reviewers agree the paper is not ready for publication.